# Consumer Liking of Turnip Cooked by Different Methods: The Influence of Sensory Profile and Consumer Bitter Taste Genotype

**DOI:** 10.3390/foods12173188

**Published:** 2023-08-24

**Authors:** Nurfarhana Diana Mohd Nor, Harshita Mullick, Xirui Zhou, Omobolanle Oloyede, Carmel Houston-Price, Kate Harvey, Lisa Methven

**Affiliations:** 1Department of Early Childhood Education, Faculty of Human Development, Sultan Idris Education University, Tanjong Malim 35900, Perak, Malaysia; farhanadiana@fpm.upsi.edu.my; 2Sensory Science Centre, Department of Food and Nutritional Sciences, University of Reading, Whiteknights, Reading RG6 6AP, UK; harshitamullick94@gmail.com (H.M.); xiruizhou123@outlook.com (X.Z.); o.oloyede@surrey.ac.uk (O.O.); 3Department of Nutrition, Food and Exercise Sciences, Dorothy Hodgkin Building, University of Surrey, Stag Hill, Guilford GU2 7XH, UK; 4School of Psychology and Clinical Language Sciences, University of Reading, Early Gate, Whiteknights, Reading RG6 6AL, UK; c.houston-price@reading.ac.uk (C.H.-P.); k.n.harvey@reading.ac.uk (K.H.)

**Keywords:** turnip, *TAS2R38*, *Brassica* vegetable, bitter

## Abstract

*Brassica* vegetables are bitter, predominantly because they contain bitter-tasting glucosinolates. Individuals with high bitter taste sensitivity are reported to have lower consumption of bitter vegetables. Studies reported that cooking methods can alter the sensory characteristics of vegetables, increasing acceptability. This study investigated consumer liking of turnip cooked by four methods (boiled-pureed, roasted, steamed-pureed and stir-fried) and related this to sensory characteristics. Additionally, this study examined the effect of the bitter taste genotype on taste perception and liking of the cooked turnip samples. Participants (n = 74) were recruited and the *TAS2R38* genotype was measured. Liking, consumption intent, perception of bitterness and sweetness of turnip were evaluated. A sensory profile of the cooked turnip variants was also determined by a trained sensory panel. There were significant differences in the overall (*p* = 0.001) and taste (*p* = 0.002) liking between cooking methods. Turnip liking was increased when preparation led to sweeter taste profiles. The *TAS2R38* genotype had a significant effect on bitter perception (*p* = 0.02) but did not significantly affect taste liking. In conclusion, the cooking method affected turnip liking, and the bitter perception in turnip was influenced by the *TAS2R38* genotype. However, taste sensitivity did not predict turnip liking in this UK adult cohort.

## 1. Introduction

Consumption of *Brassica* vegetables has been consistently shown to be beneficial to human health as they contain health-promoting compounds, including glucosinolates (GSLs) and phenolic compounds, such as flavonoids and hydroxycinnamic acids [1,2,3]. GSLs are associated with risk reduction in many kinds of cancer, such as colorectal, lung and prostate cancers [4,5]. As *Brassica* vegetables contain high levels of antioxidants, it has been claimed that they could additionally prevent other chronic diseases, such as diabetes and cardiovascular disease [6].

Despite much evidence to support the role of vegetables in health benefits, it was recently reported in the UK that vegetable intake falls short of the recommendations in both children and adults [7]. Sensory characteristics of vegetables are said to be predictors of consumer liking and consumption [8]. Bitterness in vegetables, especially *Brassica* vegetables, has been shown to be a reason for consumer rejection, while sweetness is a key influence on preference [9]. In *Brassica* vegetables, GSLs and their breakdown products contribute to sensory characteristics, where Bell et al. [10] reported that isothiocyanates (hydrolysis products of GSLs) are associated with pungency and bitterness and that intact GSLs have bitter tastes. Additionally, flavonoids are also reported to be related to bitter and astringent tastes [11]. Many interventions have been suggested to increase vegetable liking and consumption; in particular, cooking processes can alter the sensory characteristics of vegetables [12]. As reported by Zeinstra et al. [13], cooking method has an impact on vegetable liking, which is highly influenced by appearance, texture and taste. Cooking temperature causes softening of the texture; Chiang and Luo [14] reported that reducing cooking temperature and duration can maintain a pleasing appearance and texture. Baxter et al. [15] found that children’s vegetable liking depends on crunchiness and hard textures as found in stir-fried vegetables and that disliking is associated with soft and mushy textures.

Considering specifically the taste of *Brassica* on cooking, Bongoni et al. [16] found that boiling could significantly reduce the bitterness caused by GSLs, while Francisco et al. [17] reported that steaming maintained the bitterness. Poelman et al. [18] also demonstrated that boiling reduced the flavour in *Brassica* vegetables in comparison to steaming. The major factor of GSL loss is due to leaching into cooking water [19]. Boiling can also cause leaching of other taste compounds such as sugars, which then results in tasteless vegetables [20]. Roasting processes, and to some extent stir-frying, cause a Maillard reaction at high cooking temperatures, where amino acids react with sugars contributing to the formation of favourable flavours [21], although this can also result in formation of the toxin acrylamide if not controlled [22].

The ability to taste bitterness varies in humans and is related to genotype. *TAS2R38* is the gene for the T2R38 bitter receptor that is predominantly responsible for perceiving bitterness from the thiourea group in the synthetic compound 6-n-propylthiouracil (PROP) but also in the naturally occurring GSLs [23,24]. In the *TAS2R38* gene, there are three single-nucleotide polymorphisms (SNPs) (*rs713598*, *rs1726866* and *rs10246939*) that give rise to three common haplotypes within the population: PAV/PAV, PAV/AVI and AVI/AVI [25]. Individuals who carry the PAV/PAV genotype are most able to detect thiourea-containing compounds, followed by those who carry the PAV/AVI genotype, while AVI/AVI individuals have the highest detection threshold [26,27]. This was supported by Bufe et al. [23] where their studies used both PROP and phenylthiocarbamide (PTC) as bitterness markers. Furthermore, Wooding et al. [28] reported that bitter perception of goitrin (hydrolysis product of the GSL progoitrin) was also influenced by *TAS2R38* but at a weaker response than PROP and PTC. Previous studies have demonstrated that the *TAS2R28* genotype affects the perception of bitterness in *Brassica* vegetables [29,30,31]. In addition, studies have also reported that *TAS2R38* can influence intake and/or liking of other vegetables (not just *Brassica* vegetables) [32,33,34,35,36].

In summary, cooking methods are important determinants of vegetable liking as they can alter the sensory characteristics of vegetables, for example, different cooking methods can modify the perception of bitter tastes in *Brassica* vegetables. The objectives of this study were to (a) investigate consumer liking of turnip cooked using four different methods (boiled-pureed, steamed-pureed, roasted and stir-fried), (b) relate consumer liking of cooked turnips to their sensory characteristics and (c) investigate the influence of the *TAS2R38* genotype on consumers’ taste perception and liking of cooked turnip. The hypothesis was that turnip liking is influenced by cooking method, and that taste genotype would have an impact on taste perception and liking, where PAV/PAV individuals would score the bitterness intensity higher than PAV/AVI and AVI/AVI individuals, and that this would influence their liking of the samples.

## 2. Materials and Methods

### 2.1. Turnip Samples and Preparation

Fresh turnips (*Brassica rapa* subsp. *rapa*) were bought from local grocery stores in Reading. Samples were prepared in the sensory kitchen of the Department of Food and Nutritional Sciences at the University of Reading, UK. Prior to cooking, turnips were peeled, stems and tails were removed and then washed. Turnips were sliced to a thickness of approximately 0.5 cm and prepared using 4 different cooking methods: boiled-pureed, steamed-pureed, roasted and stir-fried. 

#### 2.1.1. Boiled-Pureed

A total of 1.2 L of water was added into a saucepan and heated until boiling. Then, 750 g of sliced turnips was added into the saucepan and boiled for 10 min. The turnips were then drained and blended using a hand blender (Russell Hobbs) for approximately 5 min until the texture was smooth. 

#### 2.1.2. Steamed-Pureed

A total of 750 g of sliced turnips was placed into an electric steamer (Tefal) with 1 L water added to the base of the steamer and steamed for 15 min. Turnips were then blended using a hand blender (Russell Hobbs) for approximately 5 min until the texture was smooth.

#### 2.1.3. Roasted

The oven was pre-heated to 200 °C. Sliced turnips (260 g) were placed on a baking tray and drizzled with vegetable oil (3 mL). The baking trays were then placed into the oven (2 at the front and 2 at the back of the oven) and roasted for 15 min. At 7.5 min, the 2 trays at the back were swapped to the front and vice versa. After 15 min, the turnip slices were turned over and roasted for 5 more min. Turnips that were excessively burnt were discarded. 

#### 2.1.4. Stir-Fried

A total of 3 mL of vegetable oil was poured into a cooking pan and heated. Then, 260 g of sliced turnips was added to the pan and heated whilst stirring occasionally for 7 min, until they were soft and slightly brown.

To ensure no batch-to-batch variation between consumer and trained sensory panel tasting sessions, all cooked samples were placed into plastic containers, labelled and stored frozen at −18 °C prior to testing (storage time approximately 2 to 3 weeks).

### 2.2. Sample Serving

Prior to serving, all sample types were defrosted, reheated in a microwave (800 W) and stirred every 1 min until the temperature reached >75 °C. Roasted and stir-fried turnips were served on a petri-dish while both boiled-pureed and steamed-pureed turnips were served in a 30 mL transparent polystyrene cup. All samples were labelled with 3-digit random codes. Each serving consisted of either 2 slices of roasted or stir-fried turnips or approximately 15 g of boiled- or steamed-pureed turnips. Samples were placed on heat-resistant trays and placed on a hot plate to keep them warm while serving (40–45 °C). Water and plain crackers (Carr’s table water crackers, UK) were given for palate cleansing.

### 2.3. Sensory Analysis

Sensory analysis was carried out by 10 trained panellists, each with a minimum of 6 months’ experience, using sensory profiling. The panel developed a consensus vocabulary for the 4 turnip samples, concerning aroma, flavour and taste, over 3 training sessions. During the sessions, the panel were asked to sniff and taste the samples, and reference standards (e.g., spinach, mashed potato, sucrose and quinine sulfate solutions) were used to help the panel standardise the vocabulary development. With the help of the panel leader, the terms produced were discussed and led to the consensus sensory vocabulary described in Table 1. The focus of this study was on the flavour characteristics resulting from the different cooking methods. As the samples were all reheated by microwave as outlined above, it was not appropriate to determine and score attributes related to appearance and mouthfeel. During duplicate evaluations, samples were presented monadically in a balanced sequential order and each characteristic was scored on an unstructured line (scaled 0–100), using Compusense Software (ON, Canada), except for the bitter and sweet characteristics, where a structured scale was used against the standards shown in Table 1. For bitter taste, the anchor positions were 8.1, 23.0, 38.9, 63.2 and 82.6, respectively. For sweet taste, the anchor positions were 13.8, 29.1, 57.6 and 80.6, respectively. These anchor positions were the panel mean scores for the respective reference standards (see Table 1). Evaluation sessions were conducted in a sensory room within the Sensory Science Centre at the Department of Food and Nutritional Sciences, Reading, UK. Each panellist sat in an individual booth equipped with artificial daylight and with room temperature controlled (approximately 22 °C).

### 2.4. Consumer Recruitment and Acceptability Test

This study was given a favourable opinion to proceed by the University of Reading School of Chemistry, Food and Pharmacy Research Ethics Committee (study number 14/40). Consumers were recruited from university staff and students (n = 74). Consumers gave written informed consent upon arrival and sat in an individual booth. DNA buccal swab samples were taken prior to sample tasting (Section 2.5). Consumers were asked to taste all samples and rate their liking (overall, taste, texture and appearance) using a 9-point hedonic scale (from dislike extremely (1) to like extremely (9)). Although the sensory profile had not included appearance and mouthfeel characteristics (as discussed above), it was necessary to ask consumers to rate their liking of appearance and texture in order to avoid ‘halo’ effects when considering their liking of taste and to establish the extent to which appearance and texture influenced overall liking. Consumption intent was rated using a 5-point scale (from definitely would not eat (1) to definitely would eat (5)). Individual perceptions of bitterness and sweetness of each sample were collected using a general labelled magnitude scale (gLMS). Consumers first practiced using the scale by rating their remembered perception of the sweetness of honey, bitterness of espresso, sourness of lemon and saltiness of crisps before sample tasting and scoring. The gLMS non-linear scales have descriptive anchors at a point of ‘no sensation’, ‘barely detectable’, ‘weak’, ‘moderate’, ‘strong’, ‘very strong’ to ‘strongest imaginable sensation of any kind’. The gLMS data were exponentiated and normalised for analyses to reduce scale bias effects. A normalisation factor for each consumer was derived by dividing the mean exponentiated scores across all taste perceptions (sweetness, bitterness, sourness and saltiness) from the practise gLMS scale ratings of all consumers by the mean scores for the same practise taste perceptions for each consumer. The exponentiated values of gLMS scores were then multiplied by the normalisation factor. 

### 2.5. DNA Extraction and Genotyping

Consumers were asked to swab the inside of their cheeks for approximately 1 min on each cheek using Isohelix DNA buccal swabs. These were then stored at room temperature until DNA extraction and kept dry through the use of Isohelix Dri-Capsules (Cell Projects Ltd., Kent, UK). The swabs were sent to IDna Genetics Ltd. (Norwich, UK) for extraction and genotyping, with 10% of the swabs sent as blinded replicates to ensure accuracy. DNA was extracted using Isohelix Buccalyse DNA Extraction Kit (Cell Projects, Kent, UK) according to the manufacturer’s instructions, and then diluted 1:8 with water prior to analysis. *TAS2R38* polymorphisms (*rs713598, rs1726866* and *rs10246939*) were analysed using the KASP genotyping chemistry (LGC Group, Middlesex, UK). Diluted DNA was dried into 384-well PCR plates (Life Technologies, UK), and then 5 μL of KASP Master mix (LGC Group, Middlesex, UK) and primers were added. PCR amplification was performed as follows: 94 °C for 15 min, 94 °C for 15 s, 65 °C for 20 s, 94 °C for 15 s, 57 °C for 20 s (Life Technologies, UK). The fluorescent products were detected in an Applied Biosystems instrument (Life Technologies, UK).

### 2.6. Statistical Analysis

Sensory profile data were analysed using two-way ANOVA in a mixed model where assessors were fitted as random effects and samples as main effects, and effects were tested against the assessor by sample interaction. For consumer data, one-way repeated measures ANOVA were used to compare means (liking scores, consumption intent and taste perceptions) between cooking methods. Mixed ANOVA was used to determine the interactions between cooking method and *TAS2R38*. Multiple pairwise comparisons post ANOVA were carried out using Tukey’s HSD at a significance level of 5%. Pearson’s correlation was used to determine associations between taste perception and consumer liking. Multiple linear regression was used to test the ability of taste perception to predict turnip liking. Agglomerative hierarchical cluster (AHC) analysis was used to identify groups of consumers with different liking patterns. Dissimilarity was determined by Euclidean distance and agglomeration using Ward’s method (automatic truncation). To relate consumer liking of cooked turnips to sensory characteristics, an internal preference map was created using principal component analysis (PCA). Sensory characteristics and cluster means of consumer liking were projected onto the PCA, as supplementary data. Sensory profile data analysis was carried out in SENPAQ (Qi Statistics Ltd., Reading, UK) while consumer data were analysed using XLSTAT (version 2015.6.01, Addinsoft, Paris, France).

## 3. Results

### 3.1. Sensory Characteristics of Cooked Turnip

Twenty-three characteristics associated with aroma, taste and flavour were identified. Table 2 shows the mean sensory characteristic scores for the turnip cooked using four different methods. Significant differences were found for all the aroma characteristics except for the sweetcorn and tannin aromas. The caramelisation and burnt aromas were scored significantly higher in the roasted turnip than in all the other samples. In addition, the roasted turnip had significantly higher scores for the savoury and sweet aromas than the boiled-pureed turnip. Both puree samples had a significantly higher score for the starchy aroma than the roasted and stir-fried turnip.

For taste characteristics, there were significant differences in the salty, umami and sweet tastes between the cooking methods. There was no significant difference in the bitter taste, although all the samples were recognised as bitter (Table 2). The boiled-pureed turnip was significantly lower in sweet taste than the turnips cooked using all the other methods. The increase in sweetness might be expected to supress bitterness [37]; however, only the roasted turnips had a lower mean bitter score than the turnips cooked by the other methods, and this was not significant. The umami taste was significantly higher in the roasted turnip than in the boiled- and steamed-pureed turnips. 

In terms of the flavour characteristics, the results revealed significant differences between the cooking methods in the earthy, burnt and apple flavours. The roasted turnip had a significantly lower score for the earthy flavour than the steamed-pureed turnip but was significantly higher for the burnt flavour than all the other cooking methods. 

In summary, the sensory profile from the trained panel indicated that all the cooking methods produced the same level of bitterness in the turnips; however, in terms of sweetness, the boiled-pureed turnip had the lowest score.

### 3.2. Consumer Demographics, Taste Genotype Characteristics

A total of 74 consumers participated in this study. The age range was 18 to 62 years (mean age: 27.6 years, median age: 23.0 years). As shown in Table 3, the majority of the participants were female (82.4%), and more than half were white (52.7%). Concerning *TAS2R38*, 40.5% carried the PAV/AVI genotype, 31.1% had the AVI/AVI genotype, 18.9% had the PAV/PAV genotype and 9.6% had rare genotypes. 

### 3.3. Consumer Liking and Consumption Intent of Cooked Turnips

As shown in Table 4, there were significant differences in the overall and taste liking between the cooking methods (F(3219) = 5.66, *p* = 0.001 and F(3219) = 5.02, *p* = 0.002, respectively), where the roasted turnip was significantly more liked than the boiled-pureed turnip (*p* = 0.001 and *p* = 0.002, respectively). There was also a significant difference in the liking of the texture (F(3219) = 3.88, *p* = 0.01); however, the post hoc test did not show any significant differences between the specific sample pairs. The liking of the appearance was not significantly different between the samples. A similar pattern of significant differences between the samples was found for consumption intent (F(3219) = 10.17, *p* < 0.001). The results showed that consumers were significantly more likely to consume roasted turnip than steamed-pureed (*p* = 0.01) or boiled-pureed turnips (*p* < 0.001) and significantly more likely to consume stir-fried turnip than boiled-pureed turnip (*p* = 0.01). 

### 3.4. Consumer Taste Perception of Cooked Turnip, and the Effect of Taste Genotype and Influence on Liking

In line with the sensory profile results, there was no significant difference in the bitter perception rating by consumers between the cooking methods. However, there was a trend (F(3219) = 2.38, *p* = 0.07) where the steamed-pureed turnip had the highest mean score (20.8) and the boiled-pureed turnip had the lowest mean score (16.0). For sweet perception, although a significant difference in ratings was found between the cooking methods (F(3219) = 3.59, *p* = 0.01), the post hoc tests did not reveal any significant differences between the samples. However, the roasted turnip had the highest mean score (14.9), and the boiled-pureed turnip had the lowest mean score (10.6). It is noted that these conclusions are drawn from a relatively small size (n = 74) of consumers in the UK only; it would be useful to test whether such trends were significant with a larger population size. As shown in Table 5, taste liking was negatively correlated with bitter perception but positively correlated with sweet perception. In addition, multiple linear regression was used to test if bitter and sweet perception significantly predicted taste liking for turnip. The results indicated that these two predictors explained 15.7% of the variance (adjusted R^2^ = 0.157, (F(2, 293) = 28.43, *p* < 0.001). Both bitter (B= −0.33, *p* < 0.001) and sweet perception (B = 0.20, *p* < 0.001) significantly predicted taste liking. 

The *TAS2R38* genotype had a significant effect on bitter perception (F(2256) = 4.14, *p* = 0.02); the PAV/PAV consumers tended to score higher for bitter intensity than the PAV/AVI (*p* = 0.07) and AVI/AVI consumers (*p* = 0.05) across all the samples (27.6 compared to 16.9 and 16.4, respectively). There was no significant difference in the bitter intensity score between the PAV/AVI and AVI/AVI consumers (*p* = 0.99). The interaction between the cooking method and *TAS2R38* approached significance (F(6256) = 1.96, *p* = 0.07). The PAV/PAV consumers generally scored bitter intensity more highly for all the samples; however, it was only for stir-fried turnip that the PAV/PAV consumers rated bitter intensity significantly more highly than the PAV/AVI (*p* = 0.02) and AVI/AVI (*p* = 0.001) consumers (Figure 1). The *TAS2R38* genotype had no significant effect on sweet perception (F(2256) = 2.56, *p* = 0.08), where there was no significant interaction between the cooking method and *TAS2R38* (F(6256) = 1.07, *p* = 0.38). 

Although the individuals with rare genotypes were excluded from the analyses, it is interesting to note the pattern for bitter perception in these individuals. The consumer with PAV/AAI scored bitterness most highly (34.3), followed by the consumers with AAV/AVI (21.8) and AAI/AVI (12.1) and four consumers with PAV/AAV (11.6) (all exponentiated means). However, with only seven consumers displaying the rare genotypes (9.5% of the study population), a much larger population would be needed to analyse for such trends. 

Although the *TAS2R38* genotype had a significant effect on bitter taste perception, this bitter taste genotype did not have a significant influence on taste liking of the cooked turnips (F(2, 70) = 0.99, *p* = 0.38) nor on overall liking (F(2, 70) = 1.05, *p* = 0.35); there was no significant interaction between the cooking method and *TAS2R38* for taste liking (F(6210) = 0.77, *p* = 0.59) nor for overall liking (F(6, 210) = 1.38, *p* = 0.23). 

### 3.5. Hierarchical Cluster Analysis of Consumer Liking Data

A hierarchical cluster analysis of the overall liking data showed that the consumers could be categorised into three clusters (Table 6). Cluster 1 consumers (28.3%) liked all the samples and cluster 2 (49.3%) disliked all the samples; there were no significant differences in liking between the samples within either of these clusters. The consumers in cluster 3 (22.4%) neither liked nor disliked the stir-fried turnip; these consumers liked the roasted turnip and disliked both the boiled- and steamed-pureed turnips. The proportion of consumers who were tasters (PAV/PAV and PAV/AVI) versus non-tasters (AVI/AVI) was similar to each cluster ((cluster 1: 63% tasters vs. 37% non-tasters, cluster 2: 67% tasters vs. 33% non-tasters and cluster 3: 67% tasters vs. 33% non-tasters)), concluding that the separation of these clusters is not influenced by the bitter taste genotype. 

### 3.6. Internal Preference Map

The sensory characteristics and cluster data were regressed onto a principal component analysis (PCA) of the consumer liking data to produce an internal preference map (Figure 2). The first dimension (PC1) explained 44.8% of the variation within the overall liking data, while the second dimension (PC2) explained 34.5% of the variation. The first dimension was highly correlated with the overall liking of cooked turnip for the consumers in cluster 1 (r = 0.90) and cluster 3 (r = 0.86). The consumers liked the samples that had a sweet taste (r = 0.44), caramelised aroma (r = 0.72), sweet aroma (r = 0.82), burnt aroma (r = 0.56) and burnt flavour (r = 0.52) but disliked the bitter taste (r= −0.40), earthy aroma (r= −0.96) and earthy flavour (r= −0.91). Sweet (aroma and taste) and caramelised aroma were positioned along with roasted turnip in the top right of the plot and were negatively correlated with bitter taste which was positioned in the bottom left of the plot along with steamed- and boiled-pureed turnip. The third dimension explained a further 20.7% of the variation in the data and was highly correlated with the consumers in cluster 2 (r= −0.96), who disliked all the samples.

## 4. Discussion

In this study, it was found that turnip acceptability was influenced by the cooking method. The roasted turnip had significantly higher overall and taste liking than the boiled-pureed turnip. The consumers were significantly more likely to consume the roasted turnip than the boiled-pureed turnip. The proportion of people that disliked all the samples (49.3%) was very high, which highlights the importance of increasing consumer liking of such vegetables. We acknowledge one limitation of the current study was that the samples were frozen, defrosted and re-heated before serving to ensure sample homogeneity; this will have led to changes in texture [38] that may have reduced differences due to cooking and would not be fully representative of the normal consumer experience. It is also noted that the frequency of *TAS2R38* tasters and non-tasters in each cluster was similar, suggesting that the differences in the overall liking score of the cooked turnips between the clusters is not influenced by the *TAS2R38* genotype. This is perhaps expected as the overall liking included texture and appearance liking where the bitter taste genotype would not influence these modalities. 

A negative correlation between taste liking and bitter perception and a positive correlation between taste liking and sweet perception were found. This suggests that as the bitterness increases in cooked turnip, liking decreases, but as the sweetness increases, liking increases. The results also showed that both bitter and sweet perception were significant predictors for taste liking in turnip. Similar findings were reported in previous studies where consumers’ preferences for vegetables were influenced by lower bitterness and higher sweetness [33,39,40,41]. This pattern was particularly true for the consumers in cluster 3 (22.4% of the study population), who liked the roasted turnip significantly more than the boiled- and steamed-pureed turnips, and for whom the roasted turnip was positively associated with a sweet taste and negatively associated with a bitter taste. It may also explain why these consumers rated the overall liking lower for the boiled- and steamed-pureed turnips, as these cooking methods led to bitter tasting samples that were less sweet (in the case of boiled-pureed turnip) and had more earthy and/or less sweet-associated apple flavour notes compared to the roasted and stir-fried samples. Mennella and Bobowski [42] and Schwartz et al. [43] explained that humans are born with a preference for sweet taste and a dislike of bitter taste, which aligns with the correlations between taste liking and taste perception in this study. 

As the *TAS2R38* genotype is primarily responsible for the perception of bitterness from the thiourea group in GSLs, its effects on the taste perception of cooked turnips were analysed. The results revealed that there was a significant effect of the *TAS2R38* genotype on bitter perception. The PAV/PAV consumers tended to perceive higher bitterness in the turnip than the PAV/AVI and AVI/AVI consumers, across all samples. A similar result was reported by Bell et al. [29] for bitter perception in rocket; PAV/PAV individuals perceived a higher bitter intensity than the other two *TAS2R38* genotype groups in seven cultivars of rocket. Consistent with Sandell and Breslin’s [30] findings, the PAV/PAV individuals rated *Brassica* vegetables more bitter than the AVI/AVI individuals. Rare genotypes were not included in this statistical analysis. From observation, those with PAV/AAI scored bitterness most highly, followed by AAV/AVI, AAI/AVI and PAV/AAV. Bufe et al. [23] suggested that the AAI and AAV haplotypes perceived intermediate bitterness intensity from 6-n-propylthiouracil (PROP), a synthetic compound that contains a thiourea group. However, the number of consumers who carried these rare genotypes in this study were too low to draw any firm conclusions. 

On the other hand, this study found that there was no effect of the *TAS2R38* genotype on taste liking across all the samples, and there was also no significant interaction between the cooking method and *TAS2R38* genotype. This indicates that a genetic predisposition to bitter taste does not strongly influence consumers’ liking of bitter vegetables nor influence their liking of cooking methods that modify the bitter taste, despite these having a direct influence on bitter taste perception. However, it would be useful to further validate this conclusion in a larger population.

Although sensory profiling found no significant differences in the mean scores for bitter taste between the cooking methods, the PAV/PAV consumers did rate the stir-fried turnip to be significantly more bitter than the PAV/AVI and AVI/AVI consumers. For any *Brassica* vegetable, it is important to consider the impact of cooking on myrosinase enzyme activity (the greater the inactivation, the lower the hydrolysis of GSLs), as well as leaching into the cooking water (the less leaching, the greater the retention of GSLs). In a paper evaluating the effects of three cooking methods (steaming, microwaving and stir-frying) on cabbage, Oloyede et al. [44] found myrosinase was most stable after stir-frying and resulted in the lowest GSL concentration. However, in that study the cabbage was stir-fried for only 1.5 min, whereas in the present study the turnips were stir-fried for 7 min; it is possible that as the heat exposure was much greater in the present study, the GSLs would have been retained due to both enzyme deactivation and the lack of leaching into the cooking water. Indeed, Nugrahedi et al. [19] reported that stir-frying is one of the best cooking methods for retaining GSLs in *Brassica* vegetables. This might explain the PAV/PAV consumers’ response: stir-frying may have led to a greater retention of GSLs compared to boiling or microwaving, and although it led to greater sweetness compared to boiled-pureed turnip, this was not sufficient to mask the bitterness for the bitter-sensitive consumers compared to other consumers. 

## 5. Conclusions

Consumer liking of turnip is dependent on the cooking method, with roasted turnip being the most liked and boiled-pureed turnip the least liked. The *TAS2R38* genotype had an impact on bitter perception but not on taste liking of cooked turnip. There was a tendency for the PAV/PAV consumers to perceive higher bitterness than the PAV/AVI and AVI/AVI consumers. Sweetness was found to be a driver of turnip liking, while perceived bitterness decreased liking. Considering the health benefits of consuming *Brassica* vegetables, and the fact that turnips can be readily grown in the UK, a simple but potentially impactful recommendation from this study would be to promote the inclusion of stir-fried and roasted turnip in mainstream UK diets. However, such cooking methods are favourable as the Maillard reaction generates the desirable flavours and so should be optimised to ensure any formation acrylamide can be minimised.

## Figures and Tables

**Figure 1 foods-12-03188-f001:**
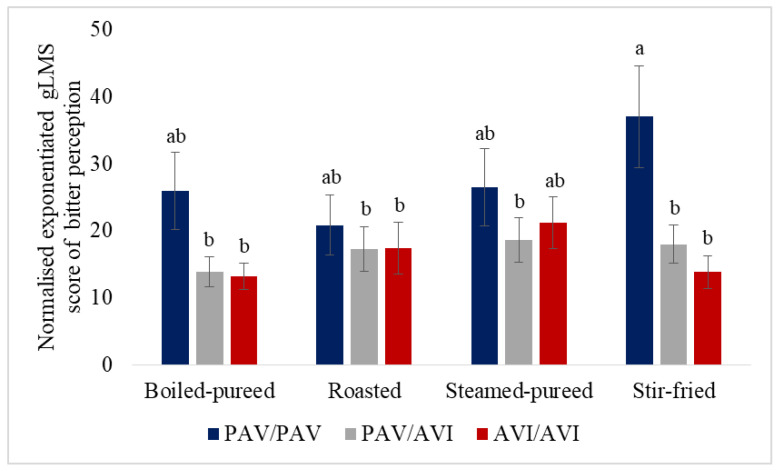
Consumer scores for bitter perception for turnip cooked using 4 different methods according to *TAS2R38* genotype. Differences in letters at the top of each bar indicate significant differences (*p* < 0.05) between any pair (genotype and cooking method) (*p* > 0.05). Values are means ± SEM.

**Figure 2 foods-12-03188-f002:**
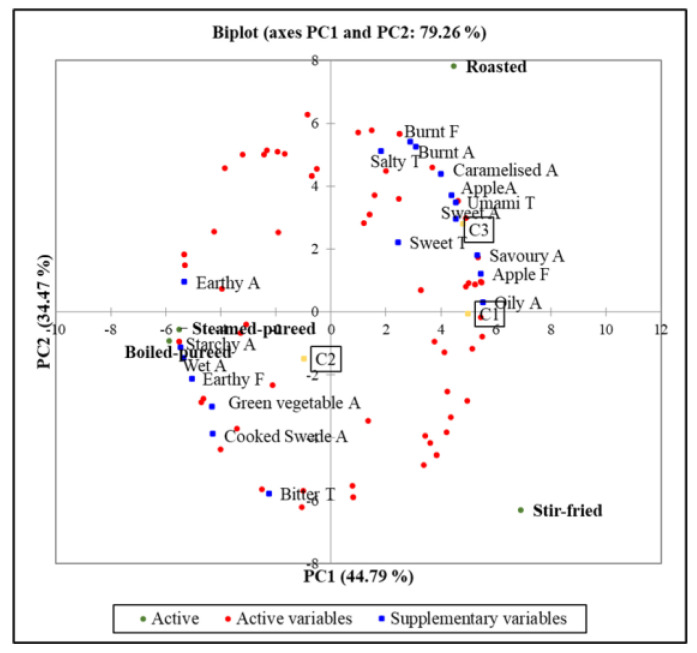
Internal preference map showing consumers’ overall liking scores (red circles) for turnip prepared using 4 cooking methods (boiled-pureed, roasted, steamed-pureed and stir-fried) with sensory characteristics (blue squares) and cluster (yellow squares) as supplementary variables. Abbreviation: A: aroma, C: cluster, F: flavour and T: taste.

**Table 1 foods-12-03188-t001:** Definition of sensory characteristics associated with samples of turnips cooked by 4 different methods and references used during vocabulary development.

Sensory Characteristic	Definition
*Aroma*	
Apple	Aroma associated with apple
Cooked swede	Aroma associated with cooked swede
Green vegetable	Aroma associated with green vegetable (spinach)
Sweetcorn	Aroma associated with sweetcorn
Savoury	Aroma associated with savoury food
Sweet	Aroma associated with sweet food
Caramelised	Aroma associated with burnt sugar
Earthy	Aroma associated with earth or soil
Starchy	Aroma associated with starchy food (mashed potato)
Tannin	Aroma associated with tea
Burnt	Aroma associated with burnt food
Wet	Aroma associated with musty
Oily	Aroma associated with cooking oil
*Taste*	
Salty	Taste associated with sodium chloride
Umami	Taste associated with monosodium glutamate
Sweet	Taste associated with sucrose solution (anchored across the scale with 0.5%, 1.0%, 2.0% and 2.6% *w*/*v* solutions)
Bitter	Taste associated with quinine sulfate solution (anchored across the scale with 0.00005%, 0.0001%, 0.0002%, 0.0004% and 0.0006% *w*/*v* solutions)
*Flavour*	
Earthy	Flavour associated with earth or soil
Tannin	Flavour associated with tea
Burnt	Flavour associated with burnt food
Green vegetable	Flavour associated with green vegetable (spinach)
Cooked onion	Flavour associated with cooked onion
Apple	Flavour associated with apple

**Table 2 foods-12-03188-t002:** Mean scores for sensory characteristics of turnips cooked using 4 different methods. Different superscript letters indicate significant differences between mean scores for cooking methods.

Sensory Characteristic	Cooking Method				Significance of Difference
	Boiled-Pureed	Steamed-Pureed	Roasted	Stir-Fried	(*p*-Value)
*Aroma*					
Apple	2.6 ^ab^	1.3 ^b^	9.2 ^a^	5.2 ^ab^	0.04
Cooked Swede	17.4 ^ab^	19.0 ^a^	8.5 ^b^	14.1 ^ab^	0.02
Green vegetable	14.7 ^ab^	19.9 ^a^	6.7 ^b^	11.6 ^ab^	0.02
Sweetcorn	4.5 ^a^	1.8 ^a^	4.0 ^a^	3.7 ^a^	0.58
Savoury	18.7 ^b^	19.4 ^ab^	27.8 ^a^	26.2 ^ab^	0.01
Sweet	14.9 ^b^	17.7 ^ab^	22.4 ^a^	19.8 ^ab^	0.04
Caramelised	0.0 ^c^	0.0 ^c^	17.4 ^a^	5.8 ^b^	<0.001
Earthy	12.6 ^ab^	14.3 ^a^	9.8 ^ab^	7.8 ^b^	0.04
Starchy	21.9 ^a^	23.3 ^a^	6.0 ^b^	6.7 ^b^	<0.001
Tannin	0.7 ^a^	0.6 ^a^	5.6 ^a^	2.1 ^a^	0.36
Burnt	0.0 ^b^	0.0 ^b^	14.1 ^a^	1.8 ^b^	0.003
Wet	20.9 ^a^	16.6 ^a^	0.4 ^b^	2.5 ^b^	<0.001
Oily	0.7 ^ab^	0.0 ^b^	6.0 ^ab^	6.8 ^a^	0.01
*Taste*					
Salty	3.0 ^b^	5.8 ^ab^	8.1 ^a^	4.1 ^ab^	0.04
Umami	15.8 ^b^	18.9 ^b^	29.8 ^a^	23.6 ^ab^	0.002
Sweet	26.9 ^b^	44.9 ^a^	45.6 ^a^	40.1 ^a^	<0.001
Bitter	26.3 ^a^	26.5 ^a^	19.3 ^a^	26.8 ^a^	0.38
*Flavour*					
Earthy	14.5 ^ab^	17.2 ^a^	6.3 ^b^	8.8 ^ab^	0.01
Tannin	5.3 ^a^	5.2 ^a^	5.2 ^a^	6.3 ^a^	0.96
Burnt	0.0 ^b^	0.0 ^b^	12.1 ^a^	1.0 ^b^	0.001
Green vegetable	14.1 ^a^	13.6 ^a^	5.2 ^a^	10.1 ^a^	0.13
Cooked onion	0.9 ^a^	0.3 ^a^	4.2 ^a^	2.8 ^a^	0.30
Apple	1.4 ^c^	2.1 ^bc^	10.8 ^a^	10.3 ^ab^	0.004

**Table 3 foods-12-03188-t003:** Demographics characteristics and taste genotype of consumers (n = 74).

Characteristic	n (%)
*Gender*	
Male	13 (17.6)
Female	61 (82.4)
*Ethnic group*	
White	39 (52.7)
Asian British	12 (16.3)
Black	5 (6.8)
Arab	3 (4.1)
Others	13 (17.6)
Preferred not to disclose	2 (2.7)
*TAS2R38*	
PAV/PAV	14 (18.9)
PAV/AVI	30 (40.5)
AVI/AVI	23 (31.1)
PAV/AAV	4 (5.4)
PAV/AAI	1 (1.4)
AAI/AVI	1 (1.4)
AAV/AVI	1 (1.4)

**Table 4 foods-12-03188-t004:** Mean liking scores (1–9) for overall, taste, texture and appearance liking, and consumption intent scores (1–5) for turnip cooked using 4 different methods. Differences in superscript letters indicate significant differences between cooking methods.

	Boiled-Pureed	Steamed-Pureed	Roasted	Stir-Fried	Significance of Difference (*p* Value)
Overall liking	4.6 ^b^	4.9 ^ab^	5.5 ^a^	5.3 ^ab^	0.001
Taste liking	4.7 ^b^	4.9 ^ab^	5.6 ^a^	5.4 ^ab^	0.002
Texture liking	4.7	4.7	5.3	5.4	0.01
Appearance liking	4.8	4.6	4.9	4.8	0.87
Consumption intent	2.4 ^c^	2.6 ^bc^	3.3 ^a^	3.1 ^ab^	<0.001

**Table 5 foods-12-03188-t005:** Correlations between taste liking and bitter perception or sweet perception (n =74).

Cooking Method	Correlation between Taste Liking and Bitter Perception		Correlation between Taste Liking and Sweet Perception	
	Pearson’s Correlation (r)	*p*-Value	Pearson’s Correlation (r)	*p*-Value
Overall	−0.35	<0.001	0.24	<0.001
Boiled-pureed	−0.14	0.24	0.24	0.04
Roasted	−0.53	<0.001	0.21	0.08
Steamed-pureed	−0.45	<0.001	0.33	0.004
Stir-fried	−0.31	0.01	0.11	0.36

**Table 6 foods-12-03188-t006:** Mean overall liking scores for 3 clusters following hierarchical cluster analysis. Different superscript letters indicate significant differences between cooking methods.

Cluster	*TAS2R38*	Cooking Method				Significance of Difference
		Boiled-Pureed	Steamed-Pureed	Roasted	Stir-Fried	(*p*-Value)
1 (N = 19)	PAV/PAV (n = 2) PAV/AVI (n = 10) AVI/AVI (n = 7)	6.2	6.5	6.7	6.8	0.23
2 (N = 33)	PAV/PAV (n = 6) PAV/AVI (n = 16) AVI/AVI (n = 11)	3.9	4.8	4.1	4.3	0.13
3 (N = 15)	PAV/PAV (n = 6) PAV/AVI (n = 4) AVI/AVI (n = 5)	3.9 ^b^	3.0 ^b^	7.0 ^a^	5.6 ^a^	<0.001

## Data Availability

The data are available from the corresponding author.

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
