# Peer review of "Consumer Liking of Turnip Cooked by Different Methods: The Influence of Sensory Profile and Consumer Bitter Taste Genotype"

_foods, 2023, doi:10.3390/foods12173188_

Round 1
Reviewer 1 Report
The authors explore whether modifiable (cooking method) and non-modifiable factors (tas2r38 genotype) affect liking of turnips in a UK sample. The question is interesting, and the approach is appropriate but there are some points that should be addressed prior to publication. Major and minor comments follow.
Major
a) Power. The number of participants is somewhat low. I don’t need a formal power analysis per se, but some of the conclusions should be tempered in this regard. That is, there is always a possibility of Type II error, as absence of evidence is not evidence of absence.
b) Missing citations – seminal work by Duffy and her students (e.g., Duffy et al 2010; Dinehart et al 2006) have been omitted, along with positive (e.g., Colare-Bento et al 2012; Smith, Estus et al 2020) and negative (e.g., Feeney et al. 2014) work on TAS2R38 on vegetables. The present work would be improved by noting this work in the introduction and contrasting it with current results in the discussion.
c) The use of normalized gLMS is justifiable but the authors need to provide more detail on how they did this.
d) Mixture suppression – one might expect bitterness to drop as sweetness increases with the different cooking techniques.
e) Did the authors consider multiple regression to complement the results in table 5 – specifically, how much variation in liking can the authors predict if they model liking as a function of both sweetness and bitterness? See Dinehart 2006 for an example.
f) Why is the cluster analysis on pg 9 called table 1 – shouldn’t it be called table 6? Also, I can’t help but wonder if the proportion of genotypes differed across the 3 clusters?
Minor
Line 75 – Consider adding some of the more foundational work here (e.g., Duffy et al. 2004; Bufe et al 2005)
Line 77 --- as well as compounds like goitrin – consider revising and citing Wooding 2010
Line 277 – why does it say ‘this is a figure’?
Line 307 – but what about overall liking? This would seem to be much more important than taste liking.
Line 351 – also Dinehart 2006.
Author Response
Reviewer 1:
The authors explore whether modifiable (cooking method) and non-modifiable factors (tas2r38 genotype) affect liking of turnips in a UK sample. The question is interesting, and the approach is appropriate but there are some points that should be addressed prior to publication. Major and minor comments follow.
Major
- The number of participants is somewhat low. I don’t need a formal power analysis per se, but some of the conclusions should be tempered in this regard. That is, there is always a possibility of Type II error, as absence of evidence is not evidence of absence.
Response: This is a very important point. We have added comments to this effect in section 3.4 (highlighted in yellow). There was already one sentence to this effect in the discussion, and a further sentence has been added there too (highlighted in yellow).
- b)Missing citations – seminal work by Duffy and her students (e.g., Duffy et al 2010; Dinehart et al 2006) have been omitted, along with positive (e.g., Colare-Bento et al 2012; Smith, Estus et al 2020) and negative (e.g., Feeney et al. 2014) work on TAS2R38 on vegetables. The present work would be improved by noting this work in the introduction and contrasting it with current results in the discussion.
Response: We have added these references in the introduction and/or discussion, please see highlighted text..
- c)The use of normalized gLMS is justifiable but the authors need to provide more detail on how they did this.
Response: We think this is already covered in detail in the methods section 2.4 where we say “ A normalisation factor for each consumer was derived by dividing the mean exponentiated scores across all taste perceptions (sweetness, bitterness, sourness, and saltiness) from the practise gLMS scale ratings of all consumers by the mean scores for the same practise taste perceptions for each consumer. The exponentiated values of gLMS scores were then multiplied by the normalisation factor.” We are not sure what else we need to add here?
- d)Mixture suppression – one might expect bitterness to drop as sweetness increases with the different cooking techniques.
Response: Indeed we agree that the increase in sweetness (found due to cooking method) would be expected to supress bitterness – however despite the Roasted turnip having a lower mean value for bitterness, this was not significant; and none of the other turnips from the other cooking methods which were sweeter than the boiled turnip, even thought they had a lower bitterness mean value. This has now been commented on in the results section 3.1 (highlighted in yellow).
- e)Did the authors consider multiple regression to complement the results in table 5 – specifically, how much variation in liking can the authors predict if they model liking as a function of both sweetness and bitterness? See Dinehart 2006 for an example.
Response: Thank you for the suggestion. The findings from multiple linear regression were added in the text (highlighted in yellow)
- f)Why is the cluster analysis on pg 9 called table 1 – shouldn’t it be called table 6? Also, I can’t help but wonder if the proportion of genotypes differed across the 3 clusters?
Response: Corrected thank you.
Minor
Line 75 – Consider adding some of the more foundational work here (e.g., Duffy et al. 2004; Bufe et al 2005)
Response: We have added the references.
Line 77 --- as well as compounds like goitrin – consider revising and citing Wooding 2010
Response: We have added the reference.
Line 277 – why does it say ‘this is a figure’?
Response: This was a formatting issue and has been corrected, thank you.
Line 307 – but what about overall liking? This would seem to be much more important than taste liking.
Response: There was no significant effect of the TASR38 genotype on overall liking either, although this would be less expected as overall liking will include texture preferences and the bitter taste genotype is unlikely to influence this. However, we have added this none-significant result to the manuscript (highlighted in yellow).
Line 351 – also Dinehart 2006.
Response: We have added the reference.

Reviewer 2 Report
This manuscript investigated consumer liking of turnip cooked by four different methods. In addition, the effect of bitter taste genotype on taste perception and liking of the cooked turnip samples was evaluated. The findings are interesting and meaningful.
While it is well-written, it presents with several grammatical errors that need reconciliation such as verb tense, overuse of acronyms which precludes readability, and minor challenges with English grammar (e.g. Line 367-375).
Author Response
Reviewer 2:
This manuscript investigated consumer liking of turnip cooked by four different methods. In addition, the effect of bitter taste genotype on taste perception and liking of the cooked turnip samples was evaluated. The findings are interesting and meaningful.
While it is well-written, it presents with several grammatical errors that need reconciliation such as verb tense, overuse of acronyms which precludes readability, and minor challenges with English grammar (e.g. Line 367-375).
Response: Many thanks for your comments, we have reviewed the grammatical errors and changes are highlighted in yellow. There was a formatting error in the discussion which had resulted in an unreadable sentence in the discussion that has now been corrected.

Reviewer 3 Report
The Authors have chosen an important topic, namely the relationship between vegetable bitterness, PROP status and the preference of the turnip samples, processed 4 different methods. The reasoning in the Introduction section is clear and logical with sufficient number of references.
I would recommend adding some references to the section mentioned bellow:
Lines 64-66: Roasting processes and, to some extent stir-frying, cause Maillard reaction at high cooking temperatures, where amino acids react with sugars contributing to the formation of favourable flavours [21]. – It might be mentioned, that simultaneously with the favourable flavour ingredients, acrylamide is also formed, which have negative effect of human health. Due to that relatively recent finding, food producers aim to achieve lower acrylamide levels with the application of lower temperatures in frying and cooking.
Lines 121-123: To ensure no batch-to-batch variation between consumer and trained sensory panel 121 tasting sessions, all cooked samples were placed into plastic containers, labelled, and stored frozen at -18 °C prior to testing (storage time approximately 2 to 3 weeks). – I accept that technique for the current study, since it provides a more homogenous sample quality for all participants. However, it would be interesting to refer to the possible effects of deep freezing and thawing on the sensory parameters. We find information about that at lines 144-146, but it would be appropriate to add a reference, or short discussion here also.
Lines 257-259: There was also a significant difference in liking of texture (F(3,219)=3.88, p=0.01); however the post-hoc test did not show any significant differences between specific sample pairs. – It would be interesting to cluster the consumer sensory data, probably there might be differences in scoring patterns. Probably after clustering in case of the more homogenous groups some significant differences would be shown in case of texture.
Lines 312-313: Cluster 1 consumers (28.4%) liked all samples and cluster 2 (48.6%) disliked all samples – it is a considerable number of participants who did not like any of the samples. In future studies how the Authors might handle the issue, that mostly such consumers would be integrated in the study, who are more open to those type of products?
Lines 406-409: Considering the health benefits of consuming Brassica vegetables, and the fact that turnips can be readily grown in the UK, a simple but potentially impactful recommendation from this study would be to promote the inclusion of stir-fried and roasted turnip in mainstream UK diets. – Based on the Authors’ findings, those methods seem to be the most accepted, where there is a good chance for the Maillard reaction. This reaction has a by-product, namely acrylamide, so both stir-frying and roasting parameters should be optimized, that the level of acrylamide can be kept low.
Technical comment:
Lines 277-278: The text remained from the template, please remove: “Figure 1. This is a figure. Schemes follow the same formatting.”
Author Response
Reviewer 3:
The Authors have chosen an important topic, namely the relationship between vegetable bitterness, PROP status and the preference of the turnip samples, processed 4 different methods. The reasoning in the Introduction section is clear and logical with sufficient number of references.
I would recommend adding some references to the section mentioned bellow:
Lines 64-66: Roasting processes and, to some extent stir-frying, cause Maillard reaction at high cooking temperatures, where amino acids react with sugars contributing to the formation of favourable flavours [21]. – It might be mentioned, that simultaneously with the favourable flavour ingredients, acrylamide is also formed, which have negative effect of human health. Due to that relatively recent finding, food producers aim to achieve lower acrylamide levels with the application of lower temperatures in frying and cooking.
Response: We have added this to the text (changes highlighted in yellow).
Lines 121-123: To ensure no batch-to-batch variation between consumer and trained sensory panel 121 tasting sessions, all cooked samples were placed into plastic containers, labelled, and stored frozen at -18 °C prior to testing (storage time approximately 2 to 3 weeks). – I accept that technique for the current study, since it provides a more homogenous sample quality for all participants. However, it would be interesting to refer to the possible effects of deep freezing and thawing on the sensory parameters. We find information about that at lines 144-146, but it would be appropriate to add a reference, or short discussion here also.
Response: Thank you for this important point, we have added discussion and reference to the discussion section (changes highlighted in yellow).
Lines 257-259: There was also a significant difference in liking of texture (F(3,219)=3.88, p=0.01); however the post-hoc test did not show any significant differences between specific sample pairs. – It would be interesting to cluster the consumer sensory data, probably there might be differences in scoring patterns. Probably after clustering in case of the more homogenous groups some significant differences would be shown in case of texture.
Response: Thank you for this very valid point. Although we agree with you from a statistical perspective, we are worried that this would lead to overinterpretation of the data because, as you noted above, the samples have been frozen and defrosted and this will have changed the texture. The main aim of the manuscript was to focus on taste, and as there is in addition sub-optimal texture resulting from the freezing/ reheating cycle, then we think we should not further (or over) interpret the texture liking data.
Lines 312-313: Cluster 1 consumers (28.4%) liked all samples and cluster 2 (48.6%) disliked all samples – it is a considerable number of participants who did not like any of the samples. In future studies how the Authors might handle the issue, that mostly such consumers would be integrated in the study, who are more open to those type of products?
Response: This is a very important point that we have now added to the discussion (highlighted in yellow). The low liking of almost half the participants is likely to be related to the freeze-reheat cycle discussed above, so we have now discussed these points together in the discussion. We do not think it would be right, for the purposes of a study such as this, to only include consumers of turnips in the study (as we might in a commercial study), because here we are interested to understand drivers of liking, and of dislike.
Lines 406-409: Considering the health benefits of consuming Brassica vegetables, and the fact that turnips can be readily grown in the UK, a simple but potentially impactful recommendation from this study would be to promote the inclusion of stir-fried and roasted turnip in mainstream UK diets. – Based on the Authors’ findings, those methods seem to be the most accepted, where there is a good chance for the Maillard reaction. This reaction has a by-product, namely acrylamide, so both stir-frying and roasting parameters should be optimized, that the level of acrylamide can be kept low.
Response: Good point! A final sentence concerning this has been added.
Technical comment:
Lines 277-278: The text remained from the template, please remove: “Figure 1. This is a figure. Schemes follow the same formatting.
Response: done, thank you.

Reviewer 4 Report
The paper on Consumer liking of turnip cooked by different methods; the influence of sensory profile and consumer bitter taste genotype is quite interesting and sheds additional light on the role of taste genotype in taste perception.
In general, the manuscript is well-written and laid down very straightforwardly, and the authors are commended for their fine work.
However, a few items are missing and the authors are asked to update the following information:
- in section 3.5 Hierarchical cluster analysis of consumer liking data, the authors must present a table with the TAS2R38 genotype frequency distribution, within each cluster.
- in section 3.6 Internal Preference Map, more specifically in Figure 2, the authors should plot the overall liking scores with different symbols according to the clusters previously identified.
The discussion should be updated to accommodate these additional findings.
Author Response
Reviewer 4:
The paper on Consumer liking of turnip cooked by different methods; the influence of sensory profile and consumer bitter taste genotype is quite interesting and sheds additional light on the role of taste genotype in taste perception.
In general, the manuscript is well-written and laid down very straightforwardly, and the authors are commended for their fine work.
However, a few items are missing and the authors are asked to update the following information:
- in section 3.5 Hierarchical cluster analysis of consumer liking data, the authors must present a table with the TAS2R38 genotype frequency distribution, within each cluster.
Response: We have added the frequency distribution of TAS2R38 in the table
- in section 3.6 Internal Preference Map, more specifically in Figure 2, the authors should plot the overall liking scores with different symbols according to the clusters previously identified.
Response: We have changed the symbols for clusters in the plot to make them clearer.
The discussion should be updated to accommodate these additional findings.
Response: We have added this discussion

Round 2
Reviewer 1 Report
No further concerns.